# Nanowire Ring Embedded in a Flexible Substrate for Local Strain Detection

**DOI:** 10.3390/ma13020347

**Published:** 2020-01-12

**Authors:** Shengkun Li, Yue Qin, Xin Li, Yuejin Zhao

**Affiliations:** 1Beijing Key Laboratory for Precision Optoelectronic Measurement Instrument and Technology, School of Optics and Photonics, Beijing Institute of Technology, Beijing 100081, China; qylxtgyx@163.com; 2Key Laboratory of Instrumentation Science and Dynamic Measurement, School of Instrument and Electronics, North University of China, Taiyuan 030051, China; qytgyx@163.com (Y.Q.); lixinxjx@163.com (X.L.)

**Keywords:** integrated optics, nanowire sensor, microring resonator, strain sensor, elastic-optic effect

## Abstract

Optical sensing has attracted more and more attention in recent years with the advance in planar waveguide fabrication processes. The photon, as a carrier of information in sensing areas, could have a better performance than electrons. We propose a novel end-to-end ring cavity to fabricate sensitive units of a strain sensor. We then propose a method of combining a flexible substrate with an end-to-end semiconductor nanowire ring cavity to fabricate novel strain sensors. We used a tuning resonant wavelength detected by a homebuilt excitation and detection system to measure applied strain. The resonant wavelength of the strain gauge was red-shift and linear tuned with increasing strain. The gauge factor was about 50, calculated through experiments and theory, and Q was 1938, with structural parameters L = 70 µm and d = 1 µm. The high sensitivity makes it possible to measure micro deformation more accurately. End-to-end coupling active nanowire waveguides eliminate the shortcomings of side by side coupling structures, which have the phasing shift with no minor optical density loss. This resonator in flexible substrates could be used not only as on-chip strain sensors for micro or nano deformation detecting but also as tunable light sources for photonic integrated circuits.

## 1. Introduction

Strain detection plays an important role in modern society, including for industrial fabrication, safety detection, scientific research, and human movement detection [1,2,3]. The strain sensors vary from the initial metal wire and semiconductor wafer to the sensitive material deposited onto a flexible substrate, with characteristics ranging from heavy and rigid to miniaturization, faster response, and higher sensitivity. The prior generation of technology has the disadvantage of being rigid and not suitable for emerging flexible, skin-attachable, or skin-like application scenarios, such as personal healthcare, human entertainment, human-machine interface, and sports performance monitoring, etc. Optical fiber has low loss, high compactness, and a large light-matter interaction area to open a new field of fiber optical sensing [4]. Optical waveguides, with a diameter of nano or micro structuration, reduce the dimensions of sensing structure and present higher sensitivity, lower power consumption, and a faster response [5]. Micro/nano waveguides are a novel candidate for precious measurement due to the strong evanescent light field, which could interact with surrounding media [6]. Semiconductor one-dimensional structures such as nanowires have interesting and unique optical properties. For instance, they provide sub-wavelength optical phenomena, a large tolerance for mechanical deformation, and as well as the advantages of feasible composition methods, uniform strength, and a glaze surface. They have been researched and applied in electronics sensing and photonics [7]. The synthesis and assembly of nanowires provided a foundation for the preparation of nanowire photonic devices [8], including temperature and humidity sensors, chemical and gas sensors, waveguides, microcavity lasers [9], and strain sensors [10]. In addition, a nanowire could be manipulated to be a ring resonator cavity through a fiber tip fixed on a manipulator [9,11]. The spectral characteristics of this ring resonator cavity show its potential to be a sensor. Polydimethylsiloxane is a common material for microfabrication due to its stable chemical properties, transparency, thermal stability, and visibility under UV (Ultraviolet) light [12,13]. 

The flexible strain sensor based on PDMS (polydimethylsiloxane) and semiconductor nanomaterials is an emerging technology that has huge application prospects due to its flexibility and ease of attachment to surfaces. A transfer-and-bond fabrication method [14] and a sandwich structure [15] show the huge application prospects of the flexible strain sensor. Our integrating strategy is to embed semiconductor nanowires into a deformable substrate. Mechanical strain applied on the integrated device then causes a change of the parameters of the sensitive element, and the subsequent spectral behavior indicates the causal association of the change of the parameters through deformation [16]. 

In this work, a photonic integrated circuit minor deformation sensor was fabricated by combining an end-to-end coupling CdS (Cadmium sulfide) based resonator cavity and a PDMS flexible and transparent substrate. Semiconductor nanowire has good mechanical and optical properties, and therefore been used here as a critical sensing element. Manipulating passive optical nanowires to function as different components such as a filter or resonator, etc. has been a mature method. Here, a CdS active semiconductor nanowire was manipulated to be an end-to-end resonator cavity through a fiber tip fixed on a precession stage. Then the manipulated cavity on a flexible substrate was put into a Paryline deposition system to package the sensing element. The fabricated sensor was put under a home-built excitation and detection system to test the performance. The detected results show a novel strain sensor based on the optical cavity.

## 2. Materials and Methods

### 2.1. Materials and Equipments

CdS was purchased from Sichuan Xinlong Tellurium Technology Development Co., LTD. (Chengdu, China).

The PDMS substrate was fabricated by mixing the PDMS solution and hardening agent in a ratio of 10:1, then stirring the mixed solution for 8 min. The mixed solution after stirring had bubbles, which can cause harm to the mechanical properties. We placed the solution in a vacuum tank for some time to remove the bubbles from the solution. We then poured the solution into the mold and heated it to solidify. We used the self-made temperature and humidity sensor to control the temperature and humidity of the experimental environment [17].

The collection and detection system was built based on an Olympus BX53 microscope (Tokyo, Japan). A Thorlabs (Newton, NJ, USA) made working distance, 50 × 0.65 N.A. objective lens was used. The CCD (Charge Coupled Device) camera was produced by Princeton Instruments (Trenton, NJ, USA), the model used was Pixis, 2K and a spectrometer with an 1800 grooves/cm grating was used.

### 2.2. Fabrication of End-to-End Ring Cavity Sensor

The CdS nanowire was fabricated using the well-known method of chemical vapor deposition [18], which provided a mature fabrication method for preparing uniform morphology, high quality nanowire. Nano metal particles were used as a catalyst to deposit the gaseous molecule and guarantee the nanowire growth along a direction. During the process of growth, the metal nanoparticles catalyze the vapor CdS precursor into a solid state within a small cross-section [19]. Silicon substrates with golden nanoparticles were put into a quartz tube oven where the CdS powder evaporated at 750 °C for 3.2 h in argon flow (99.99% purity, 90 SCCM). By conducting these operations, the CdS nanowires typically grew along [001] direction on a silicon substrate.

After the fabrication process of nanowire active waveguide and PDMS substrate was finished, the nanowire was grown directly on a silicon substrate, and a PDMS substrate showed good mechanical properties. The nanowire could be transferred to other substrates freely through a centrifuge in solution and then subsided. Figure 1a shows the fabricated nanowire waveguide on a silicon substrate. The free nanowire could be transferred to an already prepared PDMS substrate through the Van der Waal’s force involving a close attachment of two faces, as shown in Figure 1b,c. The observation under a microscope at high magnification after the move and flip of the substrate with nanowire attached on it shows the nanowire was kept still. A fiber tip fixed on a triaxial micrometer precession moving stage could be used to cleave the nanowire and manipulate the nanowire with a long working distance objective lens (50 × 0.65 N.A. Thorlabs) to form an end-to-end micro-resonator cavity, as demonstrated in Figure 1d. The three-axis precession moving stage provided the fiber tip movement in the micrometer range and avoided damage to the nanowire while bending it into a preconceived structure. After manipulating nanowire on a substrate into a proper shape, we placed the whole substrate into a Parylene vacuum vapor deposition to deposit the Parylene onto the substrate to package the CdS nanowire as shown in Figure 1e, and the the total thickness of the PDMS coating was ~1 mm. The refractive index of nanowire used in this experiment was set as n = 2.54. The diameter, cavity length, air gap length, and deviation angle were ~1 µm, ~70 µm, 300 nm and 0–60°, respectively. Figure 1f,g show the micrograph of the prepared end-to-end coupled sensitive unit and the macroscopic picture of the strain sensor, respectively. With the stretching, there were some non-ideal deformations in PDMS substrates which led to an unpredictable change in angle. Therefore, the angle fluctuated within a certain range (0–60°).

In practice, when the PDMS is stretched, the shape longitudinal variables in the middle of the resonator are larger than those on both sides. Strain will cause the two ends to sag inward, so we chose the structure with a certain outward angle in the initial state to prevent damage to the nanowire (The inward sag angle will cause nanowire fracture and other problems). Moreover, the shape of the nanowire also prevented the coupling efficiency from dropping sharply due to the change of the angle. When the strain disappeared, the PDMS substrates returned to their initial shape. Because nanowires were embedded in PDMS substrates, the nanowire also returned to its initial shape as the PDMS substrate recovered. Therefore, the sensor is reversible and reusable.

## 3. Result and Discussion

### 3.1. Principle of Laser Excitation

For semiconductor materials with direct bandgap, the material absorbed photon energy hν>E, the bandgap energy. In the experiment, a 355 nm pulse laser was used to provide high-energy photons, with an energy level of 3.50 eV. The classic method of using photoluminescence of a semiconductor with a not wide bandgap was employed to absorb a photon, then the excited carriers were relaxed through phonon emission, and the electron and hole recombined with the emission process into a photon. The relaxation process was driven by energy and momentum conservation, and the particles with positive and negative charges lost energy and momentum through energy conservation to reach the conduction band edge. Then electron and hole recombined to emit a photon, whose emitted photon energy was equal to the direct band gap energy. The whole photoluminescence experiment was conducted at room temperature, and in this circumstance, the exciton-phonon process plays a leading role. In exciton-longitudinal optical phonon scattering, when one exciton recombines, it emits a photon and one or more longitudinal optical phonon.

### 3.2. Theoretical Sensitivity Analysis

There are two main factors that cause the tuning of a wavelength: one is the change of loop cavity length, the other is the change of the nanowire refractive index. The resonant wavelength of the nanowire microring cavity can be calculated by using the formula: λ = (L/m)n. The effect of change in the nanowire waveguide cross-section dimensions caused by the strain is so small that it was neglected. Therefore, the change of the wavelength is given by:(1)$$\frac{\Delta \lambda }{\lambda }=\frac{\Delta L}{L}+\frac{\Delta n}{n}.$$

The tuning wavelength and sensitivity of the sensor could be calculated by using a change of the cavity length and nanowire refractive index with different strains applied.

The horizontal length (which contains the air gap length as shown in the insert of Figure 2a) increased and the vertical length decreased while the nanowire loop was strained, as shown in Figure 2b. As the shape of cavity under strain could be approximated as an ellipse, the length of the cavity was decided based on the loop radius of two directions. The change of cavity length is written as:(2)$$\frac{\Delta L}{L}\approx \frac{\varepsilon \left(1-\nu \right)}{2}$$
where ε is the strain and υ is the Poisson’s ratio. Since h_sub_ =~1000h_wav_, the rigidity of the substrate is much larger than the nanowire waveguide, the relationship between them is D_sub_=~10^9^D_wav_ [20,21]. Deformation of the CdS nanowire is caused by PDMS deformation. This is because the rigidity of CdS nanowires is less than that of the PDMS substrate, and deformation is applied to PDMS. Therefore, during deformation, the deformation of the CdS nanowires ring cavity in transverse and longitudinal directions should be equal to that of PDMS. So when calculating, υ is the Poisson’s ratio of the substrate. 

The wavelength was also tuned with the change of the nanowire refractive index through the elastic-optic effect. When light is propagating in the strain direction, the change of the refractive index can be written as [21]:(3)$${\left(\frac{\Delta n}{n}\right)}_{parallel}=-\frac{n^{2}\varepsilon }{2}\left[p_{12}-\nu \left(p_{11}+p_{12}\right)\right].$$

When light is propagating in the direction perpendicular to the strain direction, the change of refractive index is [21]:(4)$${\left(\frac{\Delta n}{n}\right)}_{vertical}=-\frac{n^{2}\varepsilon }{2}\left[p_{11}-2\nu p_{12}\right].$$

Therefore, the change of cavity length in theory is:(5)$$\frac{\Delta \lambda }{\lambda }=\frac{\Delta L}{L}+{\left(\frac{\Delta n}{n}\right)}_{parallel}+{\left(\frac{\Delta n}{n}\right)}_{vertical}.$$

The nanowire ring cavity sensitive unit with high sensitivity is high enough to make the sensor sense the micro or nano deformation more accurately than the previous sensor.

### 3.3. Simulation and Parameter Design

The coupling geometry of the nanowire resonator is end-to-end, and the narrowest air gap of two end facets is about 30 nm. There is a coupling angle of the two end facets that could be defined as the deviation angle between two normal nanowire ends. The nanowire coupling structure has a small calculation dimension and the finite-difference time-domain is a proper numerical method used to obtain highly accurate simulation results that could guide experimental parameter design and explain experimental phenomena. 

By using the FDTD (finite-difference time-domain) method, the coupling efficiency of light transfer between two nanowires with a deviation angle and air gap could be obtained [22]. Two cylindrical nanowires, with the same diameter of 300 nm, were placed in a central axis and the two end facets had a deviation angle set as a mathematic model. The Eigenvalue mode was guided along the input area and guided to the port of another nanowire, and the output in another nanowire was collected to calculate the coupling efficiency. Also, the sidewall of the nanowire was assumed to be smooth to simplify the calculation by ignoring the power loss due to scattering in the rough end facet. When simulating different gap lengths, the angle was 0°. When simulating different angles, the gap length was 30 nm. The simulation results show that the increase of the gap and the angle could lead to a sharp decrease in efficiency, which is shown in Figure 3c,d. Moreover, we compared nanowires of two different diameters, as shown in Figure 3b. Clearly, the 300 nm nanowires have a better ability to limit light than the 200 nm nanowires. Figure 3a demonstrates the effects of different media on coupling. When the nanowires were covered by PDMS, light coupling between the two ends improved. Since the nanowire was curved into a ring, curved loss was generated, meaning the radius of curvature was small, and thus, the curved loss increased [23]. With the unpredictable change in angle, the coupling efficiency varied. The increase in angle and curved loss lowered the quality factor.

In the experiment, the semiconductor nanowire was cleaved by using a fiber tip, and the facet after incision could be considered as smooth. The diameter of nanowire was 1 µm. Because the diameters of the two ends of nanowires were the same, the light conduction mode between the two ends could match well. Thus, the ring nanowire cavity had a perfect ability to limit light, meaning the coupling efficiency between the two ends was higher [22]. But a bigger air gap made the coupling efficiency lower. Therefore, when setting the initial geometry, a smaller air gap is better. For a single nanowire after excitation, there is an obvious spectral phenomenon of the Fabry–Perot resonator. The refractive index of air is assumed to be 1.0 and the refractive index of the CdS nanowire takes the value of 2.54. For this kind of coupling geometry, the smaller air gap length and smaller deviation angle bestowed a higher coupling efficiency. The small gap and deviation angle provided a relatively high coupling efficiency to form a resonant cavity. In addition, the nanowire sensor is sensitive to deformation and is used to measure minor deformation (maximum ε = 5%), so the deviation angle and air gap length change is small, and the coupling efficiency does not change significantly.

### 3.4. Excitation and Collection System

When we finished packaging the resonant cavity into a PDMS substrate, a homebuilt photoluminescence spectroscopy collection and detection system was built, based on an Olympus BX53 microscope. This detection system is shown in Figure 4. Due to the low transmission of the achromatic objective lens in ultraviolet band and because the high energy could harm the objective lens, a 40× silver-plated reflective objective was taken to conduct the excitation and collection of fluorescence. The total optical magnification was 400 and f. (The focal length) of camera was 150 mm. We applied a 355 nm pulse laser to excite the nanowire in the field, and a light splitting module was used to collect part of the excited light for the spectrometer to analyze the photoluminescence spectra. The excitation laser was expanded through a beam expander before being focused through the objective lens to the sample, through a reflection of the dichroic mirror. A fiber was used to collect light from the light splitting module and the core of the fiber acted as a pinhole to only collect the light from the center of the field. Light was collected from the coupling area in this experiment, the scope was about a few micrometers. A beam splitter divided the fluorescence collected into a CCD camera and spectrometer with an 1800 grooves/cm grating. Taking into account that the fluorescence spectrum changes with the power of the excitation light, the pumping power was fixed through a medium-density optical attenuator. When the laser power fluctuated within a small range, the excitation power seemed to stay steady. The fabricated sensor was put on the homebuilt testing system to verify its performance. We changed the excitation spot to achieve spectra under circumstances of different applied strains.

### 3.5. Spectrum and Properties

We observed the single nanowire placed on the surface of the silicon wafer under the microscope with white light illumination, as shown in Figure 5a. The diameter was 1 µm and the length was about 70 µm. Under the excitation of ultraviolet light, the fluorescent picture shown in Figure 5b shows three lighting spots, one is brighter and two are slightly weaker. The position of the bright spot is the excitation point, the green fluorescence difference from the ultraviolet light shows the stimulated emission of semiconductor nanowires, and the bandgap range of CdS. The slightly glowing point on both sides of a nanowire indicates a good waveguide property. Also, there were some minor spots on a nanowire, which was dust adhering on the nanowire, which leads to light intensity decreasing due to diffraction loss. Figure 5c shows the in situ (exciting and collecting at the same point) spectrum of the single nanowire where the main peak is at 524.0 nm. We found the quality factor and the full width at half maxima were 36 and 14.4 nm, respectively.

After finishing the packaging of the resonant cavity in a PDMS substrate, we put the fabricated sensor on the homebuilt testing system to verify the performance. Figure 5d shows the SEM picture of the ring cavity. The fiber collects the light from the coupling area (air gap) which is brightest, the collecting scope is about a few micrometers. A picture of a sensitive unit under UV-light excitation is shown as Figure 5e. Figure 5f shows the collected spectrum of the sensitive unit under excitation where the fluorescent image is Figure 5e, the first peak is at 518.7 nm, and the main peak is at 521.5 nm. We used the FFT (The Fast Fourier Transform) in Origin 9.1 software to filter and smooth the curve. During the smoothing of spectrum, an FFT low-pass filter was adopted. The FFT low-pass filter has a good passability for low-frequency signals. It has little influence on the peak position (wavelength at the peak position) of the spectrum, meaning it has little influence on the sensitivity. The insert of Figure 5f shows the partially magnified picture of the main peak. By analyzing the peak position, we found the quality factor of the tested resonant cavity by dividing the peak by full width at half maximum, Q = 1938. The full width at half maxima was 0.269 nm.

As shown in Figure 5c,f, the full width at half maximum of the nanowire ring cavity is much smaller than for the single nanowire cavity (F–P cavity). Therefore, the quality factor and sensitivity of the ring cavity of nanowire are both higher than that of the single nanowire cavity, which provides a new method to measure micro deformation more accurately.

### 3.6. Resonate Wavelength Tuning

The nanowire packaged in PDMS is approximately a circular shape deformed with substrate, as the diagram illustrates in Figure 2. Compared to in the simulation, the diameter of the nanowire was 1 µm. Thus, the ring nanowire cavity has a perfect ability to limit lights [22]. With the deformation of substrate, the circle becomes longer in the horizontal direction and becomes shorter in the longitudinal direction. Meanwhile, the length of the air gap increases due to the changed horizontal length. The refractive index and length of the cavity are the main reasons for the shift of wavelength. The deformation of the sensor alters the length of the cavity. We could establish a relationship between the degree of the PL spectral shift and shape change of the fabricated sensor.

Two triaxial micrometer precession moving stages were used to apply strain. Two ends of the sensor were fixed at two triaxial micrometer precession moving stages, respectively. The stretching percentage ε was defined by the elongation length of the PDMS substrate with packaging and an end-to-end coupling structure resonator cavity. Before measuring the response, we pre-pulled the sensor, and then defined the walking distance as the variable. The strain of nanowires was very small (less than or equal to 5%), and this tiny deformation can be tuned to the spectrum without much influence on the overall shape of nanowires. The photoluminescence wavelength of the CdS nanowire under excitation was related to the laser power, so the fixed pumping power and constant position of the pumping spot in the experiment avoided the wavelength tuning induced by the carrier density effect. 

The testing schematic diagram is shown in Figure 6a. The PDMS with internal sensitive units were deformed under the action of tension. After stretching, we tested the state of the sensor under the homebuilt excitation and analysis system mentioned above, and recorded the spectrum under different deformation situations. Four white light and fluorescence pictures of the cavity in different deformations are shown in Figure 6b. We extracted their peaks into a graph to obtain Figure 6c. Five spectra under different deformation situations are in the diagram. Figure 6d shows the relationship between the stretching percentage ε and the peak position.

After we measured the peak shift of the spectra to the walking distance, we obtained the following conclusion: the stretching percentage and spectral drift is a linear relationship, as shown in Figure 6d. Stretching the substrate also change the deviation angle and air gap length, then leads to a decrease in values of coupling efficiency and quality factor. Thus, the intensity of the laser decreased, with the increasing deformation and the shape of the resonance peak, and became blunt as shown in Figure 6c. We can see that the sensitivity of the ring cavity sensor from Figure 6d is about 50. As one of the main characteristics of nanowire microring cavity, the sensitivity is higher than a normal ring cavity strain sensor, the sensitivity of which is ~11.5 [24].

The theoretical sensitivity can be calculated by the parameters of the CdS nanowire ring cavity, using Equations (1)–(4). p_11_ and p_12_ in this equation are stress–optic coefficients, and they are 0.144 and 0.066, respectively [25]. The Poisson’s ratio of the PDMS substrate is 0.48. The calculation results are: ΔL/L=0.25ε, in the direction of strain Δn/n=0.1ε, in the direction perpendicular to strain Δn/n=−0.25ε, the total refractive change is the sum of refractive change in two directions Δn/n=−0.15ε, and the wavelength change and gauge factor calculation results are:(6)$$\frac{\Delta \lambda }{\lambda }=0.1\varepsilon$$
(7)$$gaugefactor=\frac{\Delta \lambda }{\varepsilon }\approx 52.$$

The shape of the loop nanowire is not exactly circular, which could influence the resonate wavelength and cause differences between the experimental sensitivity and theoretical sensitivity calculation. In addition, the uncertain deviation angle and the uncertain shape change of the ring cavity will also cause differences between the theory and experiment.

The nanowire resonator provides a way to measure micro deformation more accurately. The end-to-end coupling active nanowire waveguide eliminates the phasing shift and not the minor optical density loss of side by side coupling structure. This resonator in flexible substrates could be used not only as on-chip strain sensors for micro or nano deformation detecting, but also as tunable light sources for photonic integrated circuits.

## 4. Conclusions

In this paper, an end-to-end coupling structure resonator cavity combined with the PDMS flexible transparent substrate was fabricated to sense micro outside deformation. The end-to-end coupling structure easily achieved a stable efficiency compared to the micro loop resonator, whose coupling efficiency was not stable, and changed with the length of the coupling area and the angle at which the two ends of a nanowire waveguide cross. A finite difference time domain calculation method was used to investigate the coupling efficiency of an end-to-end coupling at different coupling structures. There is a high coupling efficiency when the two end faces have a small interspace and cross angle. The simulation result ensured the technical feasibility of fabricating end-to-end CdS nanowire resonators. The experimental results verified the simulation results: There are significant resonator peaks in the recorded spectra. In the stretching experiment, the resonator peak was sensitive to the applied strain when the calculated strain gauge factor is ~50. This provides a new way to fabricate an on-chip strain sensor.

## Figures and Tables

**Figure 1 materials-13-00347-f001:**
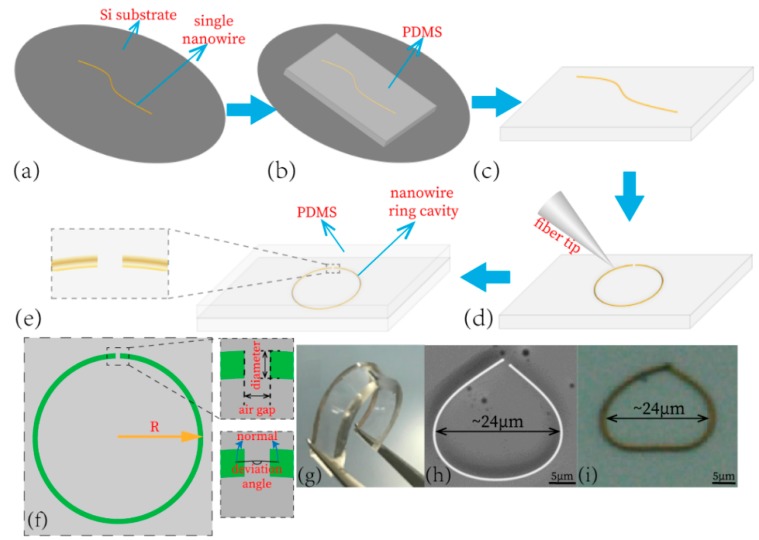
Preparation of sensitive elements. (**a**) Single CdS nanowires transferred to the surface of the silicon wafer. (**b**,**c**) The polydimethylsiloxane (PDMS) substrate is bonded to the surface of the silicon wafer, and the nanowires are transferred to the PDMS surface through intermolecular adsorption. (**d**) Manipulating nanowires with fiber optic probes to prepare sensitive elements. (**e**) PDMS substrate with sensitive cells placed in the deposition system for sensitive cell packaging. (**f**) Schematic diagram of nanowire ring cavity and its parameters. (**g**) Micrograph of the prepared end-to-end coupled sensitive unit. (**h**) Macroscopic picture of the strain sensor. (**i**) SEM (Scanning Electron Microscope) picture of the strain sensor.

**Figure 2 materials-13-00347-f002:**
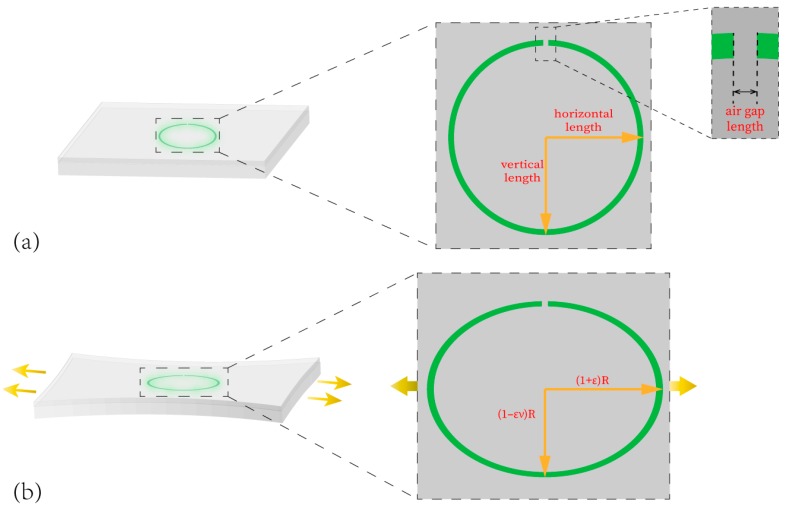
(**a**) Schematic diagram of the nanowire ring cavity. (**b**) Schematic diagram of the nanowire ring cavity under strain ε.

**Figure 3 materials-13-00347-f003:**
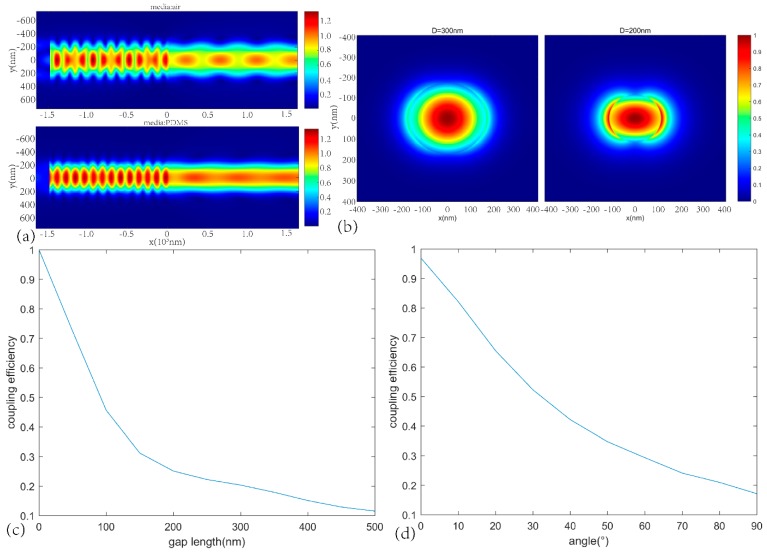
(**a**) The distribution of electric fields in different media (air above and PDMS below). (**b**) The distribution of electric fields at different nanowire diameters (the left one is 300 nm, the left one is 200 nm). (**c**) Correspondence between gap length and coupling efficiency. (**d**) Correspondence between the angle and coupling efficiency.

**Figure 4 materials-13-00347-f004:**
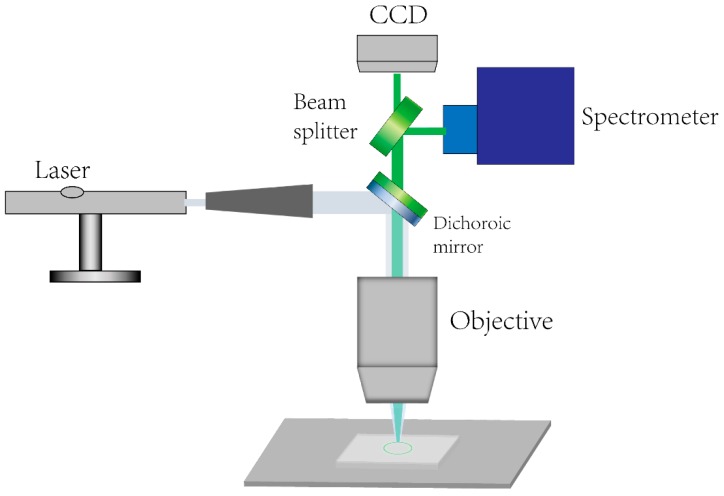
Schematic diagram of the excitation and analysis system.

**Figure 5 materials-13-00347-f005:**
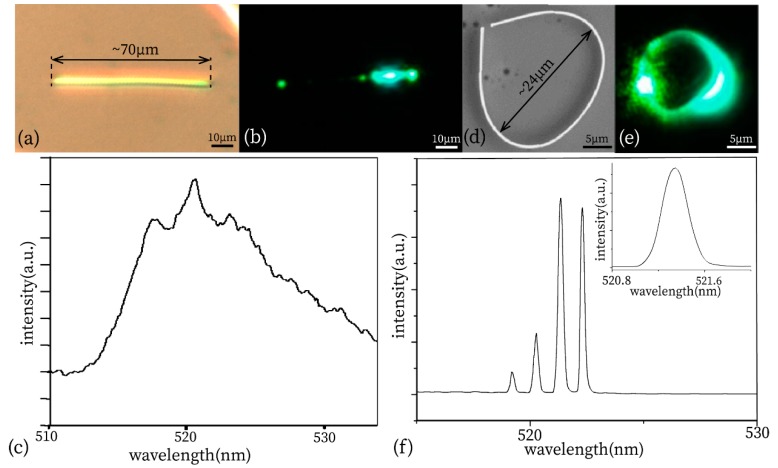
(**a**) Microscope photo under white-light illumination. (**b**) Microscope photo under UV excitation. (**c**) Spectral picture of single nanowire. (**d**) SEM picture of nanowires ring cavity under excitation. (**e**) Microscopic picture of the sensitive unit under excitation. (**f**) Spectral picture of the sensitive unit.

**Figure 6 materials-13-00347-f006:**
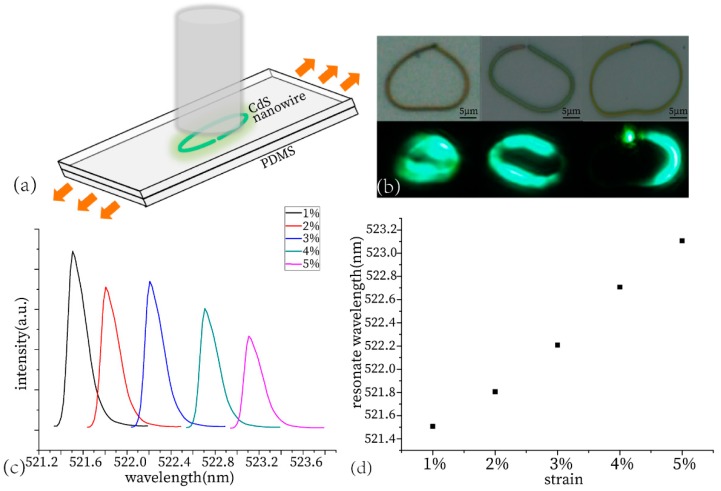
(**a**) Schematic diagram of test the packaged stress sensor. (**b**) Bright field and fluorescence picture under three different sets of deformation. (**c**) Spectral peak position map under different stresses. (**d**) Correspondence between the strain and peak positions.

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
