# Peer review of "Nanowire Ring Embedded in a Flexible Substrate for Local Strain Detection"

_materials, 2020, doi:10.3390/ma13020347_

Round 1

Reviewer 1 Report

The manuscript describes a strain sensor based on the resonance shift of a ring cavity made by a CdS nanowire placed on an elastomeric substrate.

The working principle of the device would be nice application of the resonant properties of the systems but unfortunately the article lacks of many demonstrations to be acceptable for publication.

Therefore, I suggest to reject the article for the following reasons:

First of all, in the title, the system is described as a nanowire laser while all along the manuscript any demonstration of the lasing properties of the nanowire are reported. The title should be changed or the lasing behavior of the nanowire should be reported.

The ring cavity geometrical parameters are reported at page 4. The values as the air gap and deviation angle are not relative to the reported images. If the SEM image has been acquired after the experiment this means that the sensor is not reversible and can be used only once. Reversibility issue should be discussed for sensing applications.   

For the estimation of some parameters, in the section “simulation and parameter design”, the authors use input parameters different from the ones obtained from the fabrication of the nanowires. The author should justify such discrepancies. The results of such estimation are reported in section 3.7 and not compared to the experimental results.

The analysis of experimental data is not clear and should be improved. The photoluminescence of the straight nanowire should be cleared out from the photoluminescence of the excitation point experimentally or with a data post processing to clearly show the Fabry Perot effect of the nanowire.  On the other side, the spectra reported for the ring resonator look smooth and without noise with respect to one relative to the straight nanowire. It is not clear if the authors apply some smoothening of the experimental data or how the measurement was improved. The author should also comment about the relevant losses from the resonator at the curved radius and not at the ends of the nanowire. Regarding the resonance tuning, the author should explain how the pick profile remains the same even if the mechanical deformation produces very different shapes of the ring nanowire.

Moreover, the text requires an appropriate English editing as sentences are not correct, the vocabulary is often not appropriate and many typos are present in the text. Paragraph division should also revised.

Bibliography should be expanded in order to present all the relevant results about tuning of the photonic structures on flexible substrate under strain.

Author Response

Thank you very much for your comments. We have made all required changes according to your comments and suggestions.

We change the title of our paper as “Nanowire ring embedded in a flexible substrate for local strain detected”,because our paper is about lasing properties of nanowire unless the lasing behavior.

I am sorry for my mistakes in the parameters difference between our statement and images. We change the parameters in our statements and simulations to make the parameters is relative to images(Fig.1(h) and (i)) was reported.

Reversibility was added in our paper in the end of the second paragraph of 2.2.

We change the parameters in our simulations to match our experiment and add the images of our simulations setup and results in 3.3. We also simulated the electrical distribution of nanowires coated with PDMS

The nanowire diameter is the biggest difference between simulation and experiment, thus we compare the simulation with the experiment in nanowire diameter,.

We made a statement the spectrum of single nanowire(Fig.5(C)) is in situ (exciting and collecting at the same point) spectrum which shows the Fabry Perot effect of the nanowire.

We use the FFT filter in Origin software to filter and smooth the curve.

We have added the analysis of loss from the resonator at the curved radius in simulations and experiments.

The language of our paper have been edited by MDPI Language Editing Services (ID:14871).

We expanded on some of the references about tuning of the photonic structures on flexible substrate under strain.

Thanks again for your comments on our article. If you have any comments on the revised draft, please give us your comments us again. Thank you

Sincerely yours,

Yue Qin

Reviewer 2 Report

The paper is very good. The experiment is precise, and the results are reliable. I had some small questions, but in the course of my reading the authors give very good answers

Author Response

Thanks

We have made modifications according to the opinions of other reviewers. If you have any comments, please contact us, thank you.

Sincerely yours,

Yue Qin

Reviewer 3 Report

In this work the authors provide a description of a ring shaped nanowire for a optical strain sensor. The topic is interesting and the results are sound but part of the manuscript is difficult to understand due to low quality of the english language, and I suggest to the authors to completely revise the english to make the paper more clear.

In particular

In the section "material and methods" it is not described the cvd and transfer process. It is described later in "result and discussion". This should be changed. Moreover no reference is present for the cvd used method for CdS growth.

"Simulation and parameter design" part is lacking of detail. No reference is provided for the simulation method. No graphical data is provided. This part should be improved.

The gap in the ring is simulated in the presence of air, but experimentally the nanowire is coated with Parylene. Do the oprical properties change? What is the thickness of the Parylene coating?

Figure 2 shows various images of the bent nanowire. In the picture (h) it is evident the nanowire faces don't correspond to the idealized picture (e). How this can be agree to simulations (the same happens to figure 4(d))? 

The caption of figure 2 "h) Macroscopic (Microscopic??) picture of the strain sensor. i) SEM picture of the strain sensor." seems to be inverted.

How the strain was applied in a controlled fashon to the sensor? No mechanical suetup is described.

Author Response

Thank you very much for your comments We have made all required changes according to your comments and suggestions.

The cvd and transfer is changed to section “material and methods”. And we add the references for the cvd.

We add the images and details of the simulations shown in Fig.3.

We change the simulations with the change of the parylene coating shown in Fig.3(a).

During the experiment, there were some changes in the angle and air gap, so as to achieve a better coupling effect. We supplemented them in the simulation and description in 2.2.

We have changed the position during the experiment, so it seems to have been reversed. Please do not hesitate to inform us if you think it needs to be modified.

We add the mechanical suetup in the second paragraph of 3.6.

The language of our paper have been edited by MDPI Language Editing Services (ID:14871).

Thanks again for your comments on our article. If you have any comments on the revised draft, please give us your comments us again. Thank you

Sincerely yours,

Yue Qin

Round 2

Reviewer 1 Report

As I pointed out in the first report, the paper reports inaccuracies that have not been solved yet. The author introduced some calculation detail and results but in my opinion the manuscript is not ready for publication.

The calculations are now reporting the coupling efficiency for nanowire with a diameter of 300 nm while in the SEM and optical images the diameter is 1 micron. Such calculations underline as increasing the gap length and deviation angle, the end-to-end coupling is really sensitive. In my opinion, this evidence makes the system not really suitable for a strain sensor.

Moreover, the “theoretical” calculation of the resonant wavelength shift exploits assumption as the stiffness of the CdS nanowire is 10^6 smaller than the stiffness of the PDMS substrate. The parameters are taken from Ref. 14 where they refer to a SOI platform and I do not understand the parallelism (without any further comment) with CdS nanowire on PDMS. At the same time it is not clear why the Poisson ratio of CdS nanowire should be the same than PDMS within the calculation of the refractive index change of the nanowire due to the photo-elastic effect. An estimation of the two contributes (cavity length and refractive index variation) to the resonant shift should be better done and analyzed.

To have a more clear attribution of the length cavity increase and refractive index change, the tuning of the Fabry Perot resonances of a straight nanowire can be used.

The author declare a FFT smoothing process that probably also affects the sensitivity of the sensor. If the PL signal post processing is a tool used to stabilize the data and strain detection, it should be discussed as a protocol step. Moreover, the problem of reproducibility of the tuning is neither discussed or reported with experimental data.

The field of stretchable integrated photonics is of great interest and many works report interesting results in the last decade. The author should introduce more references in the field in order to properly contextualize the work with respect to the state of the art.

Some references on :

Nanowire photonics (Pauzauskie PJ, Yang P. Nanowire photonics. Materials today. 2006 Oct 1;9(10):36-45.; Yan R, Gargas D, Yang P. Nanowire photonics. Nature photonics. 2009 Oct;3(10):569.)

stretchable photonics (Chen, Y., Li, H. and Li, M., 2012. Flexible and tunable silicon photonic circuits on plastic substrates. Scientific reports, 2, p.622.;

stretchable electronics (Amjadi M, Pichitpajongkit A, Lee S, Ryu S, Park I. Highly stretchable and sensitive strain sensor based on silver nanowire–elastomer nanocomposite. ACS nano. 2014 Apr 29;8(5):5154-63.; Kim KK, Hong S, Cho HM, Lee J, Suh YD, Ham J, Ko SH. Highly sensitive and stretchable multidimensional strain sensor with prestrained anisotropic metal nanowire percolation networks. Nano letters. 2015 Jul 7;15(8):5240-7.)

manipulation nanowires and nanowire cavity (Pauzauskie P.J., et al. Phys. Rev. Lett., 96 (2006), p. 143903)

and other relevant ones should be added.

Other remarks:

- Page 5: if the sensor is more accurate with respect to other devices, a quantification of the performances and a direct comparison (with relative references) should be reported.

- Page 3 and 9 : The author chose as initial cavity configuration with the two ends forming an “outward angle” in order to obtain an inward bending during the stretching. At page 9, the author declare that the stretching increases the deviation angle and air gap. This information should be concordant or more deeply discussed.

Author Response

Dear Editor:

We made a second revision to the paper. And we have revised the requirements of the reviewers one by one and replied to the comments of the reviewers carefully. 

We justify all the assumptions. 

For moving discrepancies between the experimental structure and theoretical and numerical analysis, we described in principle. (We referred to work of a Chinese references, but do not know whether it is appropriate.) If you think this proof process is not appropriate, please contact us,  we will redoing some computations for moving it.

Here we submit the second revised version of the paper. Please review it. Thank you very much.

Thank you and best regards.

Yours sincerely,

Yue Qin.

Round 3

Reviewer 1 Report

The authors address all the questions and comment on my previous remarks. I thus suggest to publish the manuscript in the current version after a minor revision of the English style.